# Classification of Maxillofacial Morphology by Artificial Intelligence Using Cephalometric Analysis Measurements

**DOI:** 10.3390/diagnostics13132134

**Published:** 2023-06-21

**Authors:** Akane Ueda, Cami Tussie, Sophie Kim, Yukinori Kuwajima, Shikino Matsumoto, Grace Kim, Kazuro Satoh, Shigemi Nagai

**Affiliations:** 1Division of Orthodontics, Department of Developmental Oral Health Science, School of Dentistry, Iwate Medical University, 1-3-27 Chuo-dori, Morioka 020-8505, Iwate, Japan; akane_ueda@hsdm.harvard.edu (A.U.); ykuwaji@iwate-med.ac.jp (Y.K.); shikino@iwate-med.ac.jp (S.M.); kazsatoh@iwate-med.ac.jp (K.S.); 2Department of Restorative Dentistry and Biomaterial Sciences, Harvard School of Dental Medicine, 188 Longwood Avenue, Boston, MA 02115, USA; 3DMD Candidate Class of 2025, Harvard School of Dental Medicine, 188 Longwood Avenue, Boston, MA 02115, USA; camitussie@hsdm.harvard.edu (C.T.); skim@hsdm.harvard.edu (S.K.); 4Department of Developmental Biology, Harvard School of Dental Medicine,188 Longwood Avenue, Boston, MA 02115, USA; grace_kim@hsdm.harvard.edu

**Keywords:** orthodontics, cephalograms, artificial intelligence (AI), machine learning, random forest classifier (RF), k-fold

## Abstract

The characteristics of maxillofacial morphology play a major role in orthodontic diagnosis and treatment planning. While Sassouni’s classification scheme outlines different categories of maxillofacial morphology, there is no standardized approach to assigning these classifications to patients. This study aimed to create an artificial intelligence (AI) model that uses cephalometric analysis measurements to accurately classify maxillofacial morphology, allowing for the standardization of maxillofacial morphology classification. This study used the initial cephalograms of 220 patients aged 18 years or older. Three orthodontists classified the maxillofacial morphologies of 220 patients using eight measurements as the accurate classification. Using these eight cephalometric measurement points and the subject’s gender as input features, a random forest classifier from the Python sci-kit learning package was trained and tested with a k-fold split of five to determine orthodontic classification; distinct models were created for horizontal-only, vertical-only, and combined maxillofacial morphology classification. The accuracy of the combined facial classification was 0.823 ± 0.060; for anteroposterior-only classification, the accuracy was 0.986 ± 0.011; and for the vertical-only classification, the accuracy was 0.850 ± 0.037. ANB angle had the greatest feature importance at 0.3519. The AI model created in this study accurately classified maxillofacial morphology, but it can be further improved with more learning data input.

## 1. Introduction

Digital technology in orthodontic treatment has seen rapid progression in recent years. Some of the applications of these new technologies include tracking root movement by CT scan, bending wires for orthodontic appliances using robotics, and comparison of soft tissue before and after orthognathic surgery with 3D digital simulation [1,2,3]. These novel solutions not only have exciting possibilities, they have also been found to be accurate and reliable methods for providing orthodontic treatment. 

In recent years, there has been an increasing trend of applying artificial intelligence (AI) in medical and dental fields to enhance the accuracy of diagnoses and clinical decision-making [4,5,6,7,8,9,10,11,12]. In particular, AI is a rapidly growing area in dental innovation, as seen with the large amount of AI research being conducted for various clinical applications [13,14,15,16,17,18]. An AI machine learning algorithm was used to determine tooth prognosis from electric dental records, and its accuracy was compatible with the decisions made by prosthodontists [13,15]. Another AI model was employed for a caries diagnosis, which reportedly achieved significantly higher accuracy than dentists in detecting caries lesions on bitewing radiographs, using a deep neural network [14]. In addition, a convolutional neural network was applied in designing removable partial dentures, and the accuracy of the classification of partially edentulous arches was over 99% for the maxilla and the mandible [16]. 

Machine learning was also used in the diagnostic prediction of root caries, based on the National Health and Nutrition Examination Survey data. The study showed the potential of machine learning methods in identifying previously unknown features, and the developed method demonstrated high accuracy, sensitivity, specificity, precision, and AUC in distinguishing between the presence and absence of root caries [17]. Recently, AI convolutional neural networks (CNN) have been applied in the classification of elementary oral lesions from clinical images, and the classification achieved a 95.09% accuracy [18]. Thus, AI methods have been successfully implemented in various areas of dentistry. Recent reviews discussed the great potential and challenges of AI in dentistry, and there is a clear need for trustworthy AI in dentistry [19,20,21]. 

A myriad of AI research is being conducted specifically in the field of orthodontics [22,23,24,25,26,27]. In particular, many researchers have been implementing AI in cephalogram analysis [22,23,26]. Patients presenting with dentofacial deformities often require combined orthodontic and surgical treatment, and a maxillofacial skeletal analysis is a critical component of diagnosis and treatment planning, especially when determining whether surgical intervention is necessary [22]. AI models have been found to accurately determine the need for corrective orthognathic surgery using cephalograms [22]. Additionally, AI models analyze new cephalometric X-rays at almost the same quality level as experienced human examiners [23]. These capabilities can be valuable tools for dental education and training. AI models have also been applied when determining cervical vertebrae stages for growth and development periods [26]. The timing of orthodontic treatment initiation is a crucial element of orthodontics, and it is essential to know a patient’s growth stage in order to plan the most effective treatment. Thus, accurate prediction of growth and development periods by AI could help greatly in determining treatment sequences. 

Even though digital technology can aid in selecting various treatment methods, it is still essential to formulate an appropriate diagnosis and treatment plan with a clear goal. For orthodontic diagnosis and treatment planning, each patient’s malocclusion status determines the selection of orthodontic devices and extraction patterns, if needed, for treatment. This is where the characteristics of maxillofacial morphology play a major role. Recently, cone-beam computed tomography (CBCT) has been used to obtain a three-dimensional view of maxillofacial morphological features, but it is not regularly used, due to radiation exposure [28]. Therefore, the lateral cephalogram is more commonly used in dental assessment. Lateral cephalograms are highly prevalent in orthodontic treatment planning, as they are indispensable in aiding the understanding of the morphology of malocclusion.

Various classification methods for maxillofacial morphology have been developed in the past, most of which use measurements from lateral cephalograms [29,30,31]. Among these approaches, the classification method developed by Sassouni uses cephalometric measurements to classify the maxillofacial morphology into nine types, based on the combination of anteroposterior (Classes I, II, and III) and vertical (Short-, Medium-, and Long-frame) facial types [32]. While Sassouni’s classification scheme outlines different categories of maxillofacial morphology, there is no standardized approach to assigning these classifications to patients. For instance, Sassouni did not clarify the combination of cephalometric measurements to use and the standard values for each classification [33]. As a result, currently, even among orthodontists with many years of experience, there are variations in the classifications of borderline cases. AI has proven to be an effective and accurate solution in orthodontic diagnosis and evaluation [34].

With this study, we aimed to create an AI model that uses cephalometric analysis measurements to accurately classify maxillofacial morphology. This can ensure accuracy, regardless of a practitioner’s years of clinical experience, and lead to better standardization of maxillofacial morphology classification.

## 2. Materials and Methods

### 2.1. Data Collection

The subjects of this study were Japanese males and females, aged 18 years or older, who underwent orthodontic treatment at the Iwate Medical University, School of Dental Medicine, Uchimaru Dental Center between August 2008 and June 2022. Data was collected with the following exclusion criteria: (1) patients who already underwent orthodontic intervention at the first visit, (2) patients with congenital diseases that affect maxillofacial growth, such as cleft lip and palate and chromosomal abnormalities, and (3) patients with prosthetic devices or missing teeth. Patients with jaw deformities were included. This study was approved by the Institutional Review Board at Iwate Medical University, School of Dental Medicine (approval number 01373).

### 2.2. Determining Training and Testing Data

Tracing paper was placed on each X-ray film of the 220 cephalograms to create a tracing diagram. The tracing diagrams were then imported with a scanner (GT-X900, Seiko Epson Co. Ltd., Suwa, Japan) at 100 dpi resolution. The analysis of the tracings was performed with the software WinCeph Version 11 (Rise Co. Ltd., Sendai, Japan), in which 127 measurements were calculated for each patient. Calibration settings were established by drawing a distance measurement scale on the tracing paper and using a scanner to capture the measurements of two points on the scale. From these measurements, one anteroposterior input measurement (ANB angle) and seven vertical input measurements (mandibular plane to FH, mandibular plane to SN, ramus plane to FH, ramus plane to SN, gonial angle, N-Me/Cd-Go, and overbite) were considered for analysis, illustrated in Figure 1. Three orthodontists certified by the Japanese Orthodontic Society used these eight measurements to categorize the maxillofacial morphology of each of the 220 patients into one of three anteroposterior classifications (Classes I, II, or III) and one of three vertical classifications (Short-, Medium-, or Long-frame), which placed each patient into one of nine facial classifications combining three anteroposterior classifications and three vertical classifications (Class I and Short-frame, Class II and Short-frame, Class III and Short-frame, Class I and Medium-frame, Class II and Medium-frame, Class III and Medium-frame, Class I and Long-frame, Class II and Long-frame, and Class III and Long-frame). Each orthodontist individually assessed patient data in a blinded manner. In cases where there was a discrepancy between two of the classifications for a patient, the final classification was determined by consensus between the two orthodontists who made the different initial classifications. Patients who were classified differently by all three of the orthodontists were assigned to a classification with measurements in the middle, which were class I and medium. To check inter-rater reliability, each of the three orthodontists classified patients three different times, separated by time. The facial classification determined by the orthodontists was set as an accurate classification for each patient.

A k-fold split of 5 was used to train and test each model. The 220 subjects were randomly divided into 5 folds, or groups. For each iteration, 4 groups were used to train the model, while 1 row was used to evaluate the model performance. 

### 2.3. AI Model Creation

For the combined facial classification model (with nine possible outputs), three distinct ML models within the scikit-learn (sklearn) package in Python (random forest classifier (RF), logistic regression (LR), and support vector classification (SVC)) were trained, tested, and compared in the evaluation of the cephalometric classification. The most successful classification model was used for further analysis. The best classification model used gender and the 8 cephalometric measurements to determine the combined facial classification for each patient in the training group. The accuracy score of the classification was determined for the training model by comparing the deduced AI combined facial classifications to the accurate classifications determined by the certified orthodontists. Supervised AI classification was also conducted separately for anteroposterior (Classes I, II, and III) and vertical (Short-, Medium-, and Long-frame) classifications. Anteroposterior and vertical cephalometric inputs were used for each model, respectively (Figure 2). Training and testing were performed 5 separate times, with outcome measures calculated at each iteration. The outcome measures were then averaged to find the overall model performance, and the accuracy was compared to the separate anteroposterior and vertical classifications of the accurate, combined facial classification.

### 2.4. Feature Selection

The feature importance function from the Python sklearn package was used to acquire and graph the features of greatest importance in determining the output of the combined facial classification model.

## 3. Results

### 3.1. Cephalogram Analysis Results

Cephalograms from 220 patients were collected and used in this study. The average age was 23.3 ± 5.4, and 110 females and 110 males were involved.

Analysis of the cephalograms of 220 patients by 3 orthodontists resulted in the following classification frequencies: Class I and Short 34, Class I and Medium 49, Class I and Long 9, Class II and Short 19, Class II and Medium 37, Class II and Long 14, and Class III and Short. There were 23 patients, 28 Class III and Mediums, and 7 Class III and Longs.

### 3.2. Comparing ML Models

For the combined facial classification, three distinct ML models within the sklearn package in Python (RF, LR, and SVC) were trained and tested to determine the best performing model with which to pursue analysis.

RF performed with an accuracy of 0.823 ± 0.060, an F1 score (harmonic mean of precision and recall) of 0.806 ± 0.076, recall of 0.790 ± 0.077, and a precision of 0.865 ± 0.072 (Table 1). The RF classifier performed the best across all metrics when compared to LR and SVC. Thus, the RF model was used for further analysis.

### 3.3. Nine Maxillofacial Classifications

The RF classifier was evaluated for precision, recall, and F1 score for each combined facial classification, as shown in Table 2. The highest precision was seen with the Class I and Long-frame and Class III and Long-frame outputs, and the lowest was seen with the Class I and Medium-frame output. The highest recall scores were seen with the Class II and Medium-frame classification, and the lowest was with the Class I and Long-frame classification. The highest F1 score was seen with the Class II and Medium-frame classification, and the lowest was with Class II and Short-frame classification.

There were 39 patients who were misclassified (Figure 3). Of these, 2 were misclassified in both the anteroposterior and vertical classifications, 5 were misclassified in the anteroposterior classification, and 32 were misclassified in the vertical classification. The model’s main errors came from believing that true Class I and Short-frame and Class II and Short-frame classifications were, instead, Class I and Medium-frame (20.5%) and Class II and Medium-frame, respectively (15.4%).

### 3.4. Separate Anteroposterior- and Vertical-Only Classifications

The separate anteroposterior and vertical classifications showed greater accuracy, more notably with the anteroposterior-only classification. The RF classifier was evaluated for precision, recall, F1 score, and PPV for each anteroposterior classification and vertical classification, as shown in Table 3.

#### 3.4.1. Anteroposterior-Only Classification

For the anteroposterior-only classification (Classes I, II, and III), the accuracy score was 0.986 ± 0.011 for the testing data. As visualized in the confusion matrix in Figure 4, only three anteroposterior cases were misclassified, and no specific trend was observed. Further analysis of output-specific metrics for horizontal classifications can be found in Table 4.

#### 3.4.2. Vertical-Only Classification

For the vertical-only classification (Short-, Medium-, and Long-frame), the accuracy score of the model was 0.850 ± 0.037. A total of 33 of the 220 subjects in the test group were misclassified. For the vertical classification model, the most common classification was Short misclassified as Medium, followed by Medium as Short. (Figure 5).

Further analysis of output-specific metrics for vertical classifications can be found in Table 5.

### 3.5. Feature Selection

To find which input features had the largest influence within the model, we then conducted an analysis of importance to determine which inputs had the largest importance in the nine-output combined facial classification model (Figure 6). The ANB angle had the largest influence on the model’s prediction, with a feature importance of 0.3519. The mandibular pl to FH angle followed behind, with a feature importance of 0.1691. Gender had the least feature importance at 0.0092.

## 4. Discussion

This study developed ML algorithms for classifying maxillofacial morphology using three distinct models: RF, LR, and SVC. The accuracies of the three approaches were compared, and it was found that the RF-based model was the most successful of the three. Each model uses a different algorithm of prediction: RF combines distinct, individual decision trees that each look at randomly selected feature data. The RF model was proposed by Leo Breiman in 2001 and has been commonly implemented in recent years, due to its high prediction performance [35,36]. RF is widely recognized as the most versatile model, capable of handling large amounts of data while requiring minimal pre-processing. LR computes the sum of the input features and calculates the probability of a binary occurring. Logistic regression is a popular technique used in machine learning to solve binary classification problems [37]. SVC places data on a feature space and from there, finds decision boundaries that physically separate this data. SVC has been widely applied in the field of medical image processing [38]. 

The anteroposterior classification was highly accurate, with only three misclassified. For the anteroposterior classification, the ANB angle played an important role; therefore, these three misclassified cases could likely have an ANB angle at the borderline of different classifications. In general, there are also variations in how orthodontists classify cases by ANB angle. In anteroposterior classification, cases with ANB angles > 4.0 or >5.0 are classified as Class II, and cases with ANB angles < 2.0, <1.0, or <0.0 are classified as Class III [39,40,41,42,43]. Because the AI determined ANB norms from the training data of only 220 participants in this study, the SDs determined by the AI might have been larger than the actual norm. Therefore, it is likely that there was an excess of borderline cases. Thus, the accuracy of the classifications of patients on the borderline could be improved, suggesting that more training data might be needed to create a more accurate AI model.

The accuracy of the vertical classification was relatively lower than the anteroposterior classification, as 33 cases were misclassified. Similar to the anteroposterior classification, where the training data resulted in a high SD, in the vertical classification, the low and high boundaries for the Medium classification were, respectively, lower and higher than those of the orthodontists’ evaluations. This led to a large number of cases being classified as Medium instead of Long or Short. Because the majority of the misclassified patients were classified into Medium, a possible explanation is that the Medium borderline values were lower and higher than that of the orthodontists’ range. The orthodontists’ range might not have translated into the AI model, due to not having enough training data. In addition, while the anteroposterior dimension was determined by the single measurement of ANB, the vertical dimension was classified using seven cephalometric measurements. The use of several measurements was necessary because it was impossible to classify patients with only one cephalometric measurement [44,45]. As seen in Figure 6, the feature importance was different for each measurement, which was determined by the training data. Thus, these weighted measurements might have influenced misclassification into the Medium group and thus also played a role in the accuracy of the combined maxillofacial classification with nine categories.

For the combined maxillofacial classifications, 39 patients were misclassified. Of these patients, 32 were misclassified in the vertical classification, 5 were misclassified in the anteroposterior classification, and only 2 were misclassified in both the anteroposterior and vertical classifications. In total, 34 cases were misclassified in both combined maxillofacial and vertical only classifications: 8 Mediums classified as Shorts, 6 Longs classified as Mediums, 2 Mediums classified as Longs, and 18 Shorts classified as Mediums. Thus, the vertical component played an influential role in determining the nine-category combined classifications. These 34 misclassified cases were divided into 2 groups: the cases with mandibular rotation and the cases with the borderline angle measurement of the mandibular pl to FH. Mandibular rotation included both forward and backward rotation. In the cases with mandibular rotation, the value of the ramus pl clearly deviated from the standard value; thus, the classification result might have been strongly influenced by this value. However, in cases that should be clearly classified as Long, the mandibular pl measurement, which had higher feature importance than ramus pl, also deviated from the standard value.

Therefore, the AI model was less accurate for cases involving the rotation of the mandible. The morphology of the mandibular bone is highly important in the classification of facial morphology in adults, and the mandibular plane is also said to be more susceptible to the inclination, opening, and rotation of the skull base (SN plane) [46]. Despite considering ramus pl to SN and Ramus pl to FH measurements in the AI model, the accuracy was diminished when mandibular rotation was involved. Thus, if one cephalometric measurement carried greater weight in the classification, correct classification might be more difficult for the model to achieve. For these reasons, the accuracy of the vertical classification was inferior to that of the anteroposterior classification. It may be necessary to consider additional cephalometric measurements to evaluate the degree of mandibular rotation for the AI model.

In this study, feature importance for the combined nine-category facial classification was determined, and inputs with higher feature importance included the ANB angle, the mandibular pl, and ramus pl to FH and SN; these features represent mandibular rotation and the relative anteroposterior position between the maxilla and the mandible. Interestingly, gender was found to be the least impactful feature. This indicates that the AI model created was affected by gender less than the cephalometric measurements. It was reported that there was a significant difference in the Cd-Go dimension (mandibular ramus height) between genders [47]. Therefore, we created the N-Me/Cd-Go index, which could eliminate gender differences in the mandibular ramus length. In this study, one length ratio (N-Me/Cd-Go) and seven angle measurements were used in consideration of the differences in size between individuals. This particular feature selection could lead to the creation of an AI model that is not greatly affected by gender.

The main limitation of this study is the numbers of cases used for machine learning data. This is because what we used was past data from actual practice. In addition, the use of strict exclusion criteria reduced the number of cases considerably. However, with the use of AI, we now have the potential to classify maxillofacial morphology with greater accuracy, which, until now, has been performed manually. To the best of our knowledge, this is the first report of its kind. Further analysis will be conducted with more data to improve the accuracy. Although the AI model developed in this study requires further refinement, its machine learning ability allows for the development of an accurate classification if given a substantial amount of training data. Thus, the program’s potential for accuracy and possible integration into 3D cephalometric measurements underscores its relevance in modern digital dentistry and orthodontics.

## 5. Conclusions

The AI model created in this study accurately classified patients into one of the nine combined facial classifications, based on anteroposterior maxillofacial morphology and vertical dimension. This can ensure accuracy regardless of a practitioner’s years of clinical experience and thus lead to a better standardization of maxillofacial morphology classification. This novel machine learning approach can be further enhanced with the incorporation of additional learning data. This pioneering study introduces a new potential tool to be used in future maxillofacial morphology classifications.

## Figures and Tables

**Figure 1 diagnostics-13-02134-f001:**
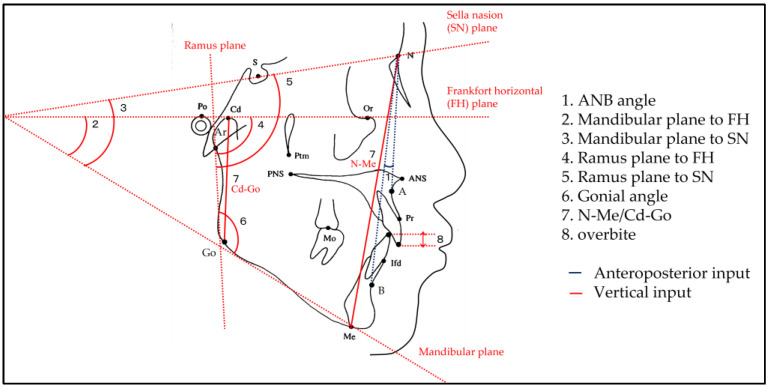
Illustration of cephalometric measurements considered for classification. Each cephalometric measurement is shown. The blue line (1) is an illustration of the measurement used in anteroposterior analysis, and the red lines (2–8) are illustrations of the measurements used in vertical analysis.

**Figure 2 diagnostics-13-02134-f002:**
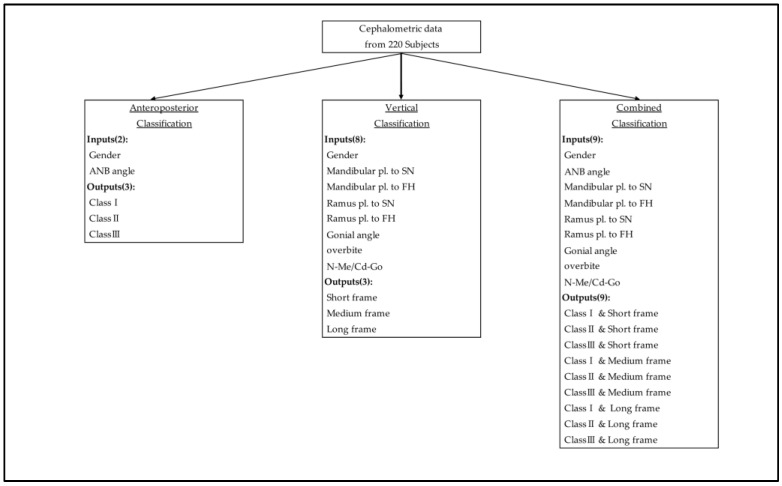
Cephalometric inputs for the determination of anteroposterior, vertical, and combined facial classifications. The input data and output classifications are listed in each category.

**Figure 3 diagnostics-13-02134-f003:**
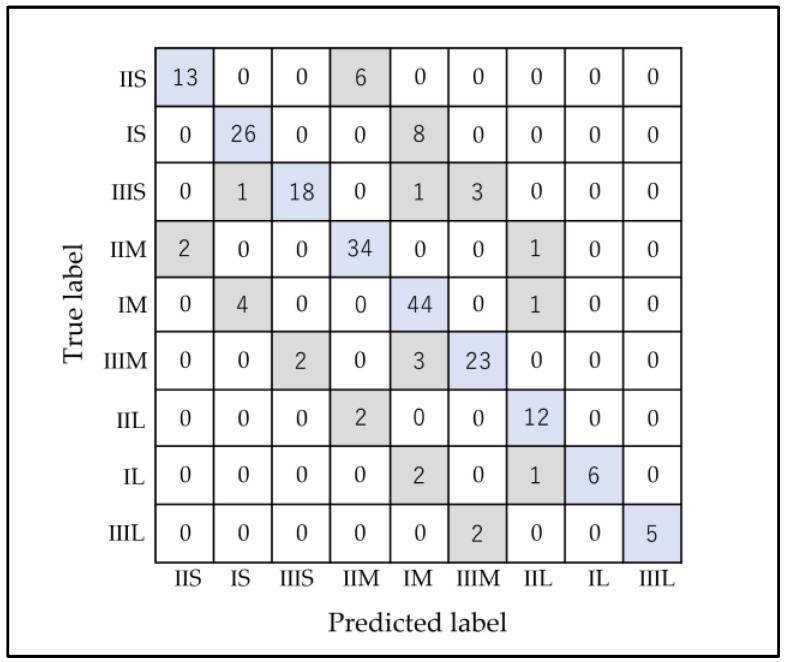
Confusion matrix for 5 runs of the combined facial classification RF. The horizontal axis represents the AI classification result (Predicted label), and the vertical axis represents the accurate classification (True label). The numbers I, II, and III represent Class I, Class II, and Class III, respectively, and S, M, and L represent Short, Medium, and Long, respectively. Blue represents patients who were classified correctly, and gray represents the misclassified patients.

**Figure 4 diagnostics-13-02134-f004:**
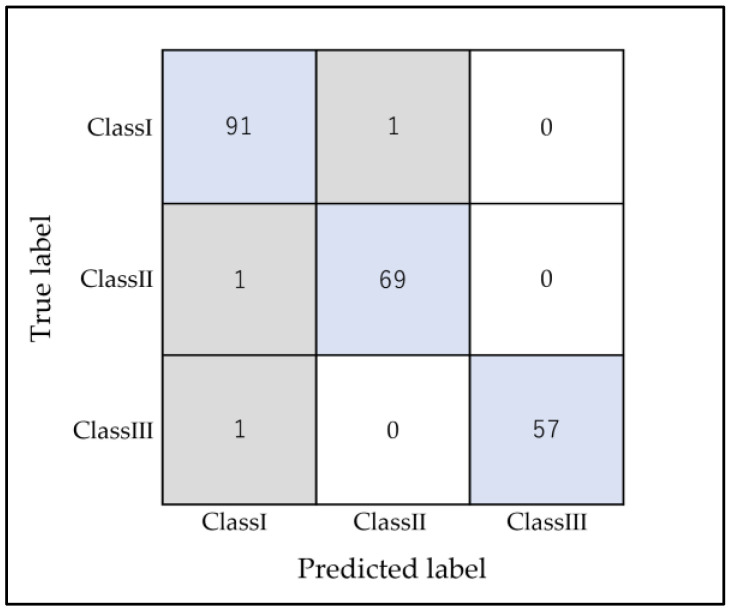
RF confusion matrix anteroposterior classifications using k-fold (n = 5). The horizontal axis represents the AI classification results (predicted label), and the vertical axis represents the accurate classifications (true label). Blue represents patients who were classified correctly, and gray represents the misclassified patients.

**Figure 5 diagnostics-13-02134-f005:**
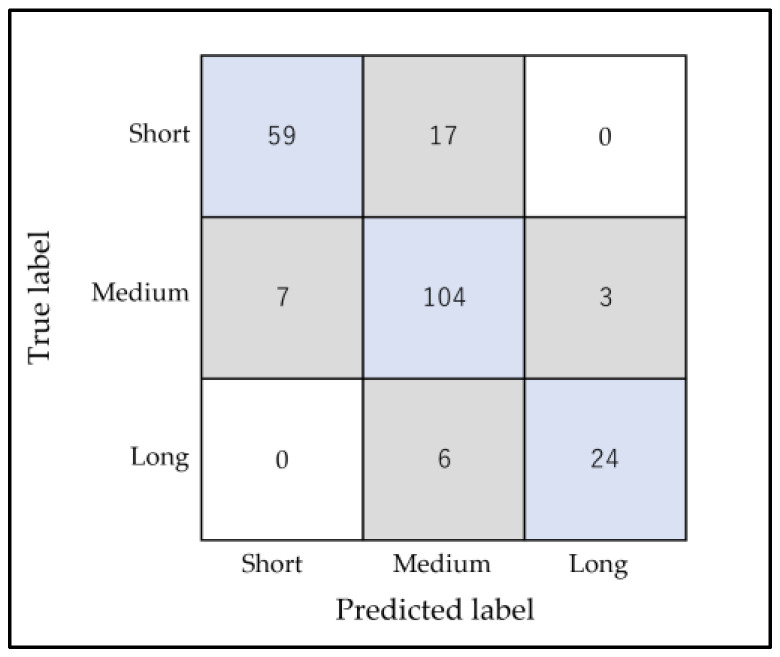
RF confusion matrix vertical classifications using k-fold (n = 5). The horizontal axis represents the AI classification results (Predicted label), and the vertical axis represents the accurate classification (True label). Blue represents patients who were correctly classified, and gray represents the misclassified patients.

**Figure 6 diagnostics-13-02134-f006:**
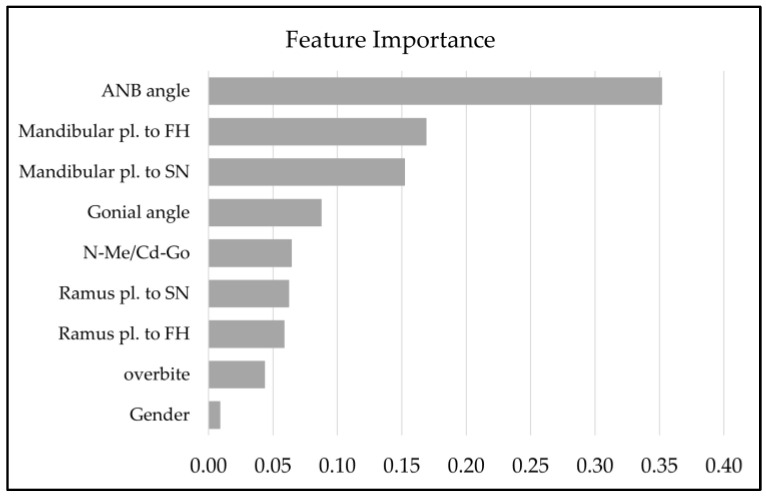
Feature importance for the combined facial classification RF. A total of 9 features were shown for gender, and 8 cephalometric items were used for classification.

**Table 1 diagnostics-13-02134-t001:** Comparison of the random forest classifier (RF), logistic regression (LR), and support vector classification (SVC) models for the 9 output classifications using k-fold (n = 5).

	RF	LR	SVC
Accuracy	0.823 ± 0.060	0.732 ± 0.046	0.677 ± 0.049
F1 score	0.806 ± 0.076	0.654 ± 0.044	0.568 ± 0.086
Recall	0.790 ± 0.077	0.660 ± 0.031	0.562 ± 0.072
Precision	0.865 ± 0.072	0.692 ± 0.050	0.629 ± 0.109

**Table 2 diagnostics-13-02134-t002:** Metrics for each combined facial classification, compared to the accurate classification RF model by classification over 5 runs.

Classification	Precision	Recall	F1 Score
Class I and Short	0.84	0.76	0.80
Class I and Medium	0.76	0.90	0.82
Class I and Long	1.00	0.67	0.80
Class II and Short	0.87	0.68	0.76
Class II and Medium	0.81	0.92	0.86
Class II and Long	0.80	0.86	0.83
Class III and Short	0.90	0.78	0.84
Class III and Medium	0.82	0.82	0.82
Class III and Long	1.00	0.71	0.83
accuracy			0.82
macro avg.	0.87	0.79	0.82
weighted avg.	0.83	0.82	0.82

**Table 3 diagnostics-13-02134-t003:** Metrics for anteroposterior classification and vertical classification RF models.

	Anteroposterior Model	Vertical Model
Accuracy	0.986 ± 0.011	0.850 ± 0.037
F1 score	0.987 ± 0.011	0.844 ± 0.035
Recall	0.985 ± 0.012	0.828 ± 0.040
Precision	0.989 ± 0.009	0.885 ± 0.044

**Table 4 diagnostics-13-02134-t004:** Metrics for the anteroposterior RF model by output.

Classification	Precision	Recall	F1 Score
Class I	0.98	0.99	0.98
Class II	0.99	0.99	0.99
Class III	1.00	0.98	0.99
Accuracy			0.99
Macro avg.	0.99	0.99	0.99
Weighted avg.	0.99	0.99	0.99

**Table 5 diagnostics-13-02134-t005:** Metrics for the vertical RF model by output.

Classification	Precision	Recall	F1 Score
Long	0.89	0.80	0.84
Medium	0.82	0.91	0.86
Short	0.89	0.78	0.83
Accuracy			0.85
Macro avg.	0.87	0.83	0.85
Weighted avg.	0.85	0.85	0.85

## Data Availability

Not applicable.

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
