# Peer review of "Classification of Maxillofacial Morphology by Artificial Intelligence Using Cephalometric Analysis Measurements"

_diagnostics, 2023, doi:10.3390/diagnostics13132134_

Round 1

Reviewer 1 Report

In this paper, the authors used the initial cephalograms of 220 patients aged 18 years or older. Three 18 orthodontists classified the maxillofacial morphology of 220 patients using eight measurements, to be considered 19 the accurate classification. There must be major revision before consideration for publication as given below:

The abstract lacks clarity in explaining the specific objectives and contributions of the research. It should clearly state the problem being addressed, the methodology used, and the key findings.

The introduction fails to provide a comprehensive background and context for the research topic. It lacks a clear justification for the need to classify maxillofacial morphology using artificial intelligence and cephalometric analysis measurement.

The literature review section is superficial and lacks depth. It fails to adequately review and critically analyze existing studies on maxillofacial morphology classification and the use of artificial intelligence. It should provide a more thorough evaluation of the limitations and gaps in previous research.

Dataset size is so small. It is required to change the dataset and reconsider large dataset.

The methodology section lacks sufficient detail and clarity regarding the cephalometric analysis measurement and the specific artificial intelligence techniques employed. It should include more specific information on the data collection process, feature extraction methods, and the training/validation procedures for the classification model.

Section 2.3. AI model creation is totally confusing and no steps are mentioned in which AI model is being created.

The section 3 results  lacks proper statistical analysis and evaluation of the classification model's performance. It should include metrics such as accuracy, precision, recall, and F1-score to assess the effectiveness of the proposed approach.

The discussion section is weak and lacks meaningful insights. It does not provide a comprehensive analysis of the results or discuss the implications and limitations of the research findings. It should provide a more critical evaluation of the proposed model and compare it to existing methods.

The conclusion does not effectively summarize the key findings and contributions of the research. It fails to address the limitations of the study and does not provide clear suggestions for future research directions.

The paper lacks visual aids or illustrations to support the presentation of results and enhance the understanding of the proposed classification model.

The organization and structure of the paper need improvement. It lacks a logical flow, and some sections appear disjointed. It should follow a clear structure with subsections to improve readability.

The language used in the paper is often unclear and convoluted, making it difficult to understand the research methodology and findings. The paper should be revised for clarity and conciseness.

Overall, this paper requires significant revisions and improvements in various aspects, including the clarity of the abstract, the depth of the literature review, the methodology details, the statistical analysis of results, the quality of the discussion, and the organization and language used throughout the paper. Addressing these issues will significantly enhance the overall quality and impact of the research.

There are some typing errors.

Author Response

Reviewer 1 : (also can check the reply in the attachment)

In this paper, the authors used the initial cephalograms of 220 patients aged 18 years or older. Three 18 orthodontists classified the maxillofacial morphology of 220 patients using eight measurements, to be considered 19 the accurate classification. There must be major revision before consideration for publication as given below:

[Response]

We are grateful to reviewer 1 for the critical comments and valuable suggestions that have helped us to improve our manuscript. As indicated in the responses described as follows, we have taken all these comments and suggestions into account in the revised version of our article.

⦁The abstract lacks clarity in explaining the specific objectives and contributions of the research. It should clearly state the problem being addressed, the methodology used, and the key findings.

[Response] Thank you very much for your feedback. We made changes to the abstract in response to your concerns.

Revised abstract

The characteristics of maxillofacial morphology play a major role in orthodontic diagnosis and treatment planning. While Sassouni’s classification scheme outlines different categories of maxillofacial morphology, there is no standardized approach to assigning these classifications to patients. This study aimed to create an artificial intelligence (AI) model that uses cephalometric analysis measurements to accurately classify maxillofacial morphology, allowing for standardization of maxillofacial morphology classification. This study used the initial cephalograms of 220 patients aged 18 years or older. Three orthodontists classified the maxillofacial morphology of 220 patients using eight measurements, to be considered the accurate classification. Using these eight cephalometric measurement points and the subject’s gender as input features, a Random Forest Classifier from the Python sci-kit learning package was trained and tested with a k-fold split of 5 to determine orthodontic classification; distinct models were created for horizontal only, vertical only, and combined maxillofacial morphology classification. The accuracy of the combined facial classification was 0.823±0.060; for anteroposterior only was 0.986±0.011; and for vertical only classification, the accuracy was 0.850±0.037. ANB angle had the greatest feature importance at 0.3519. The AI model created in this study accurately classified maxillofacial morphology, but can be further improved with more learning data input.

⦁ The introduction fails to provide a comprehensive background and context for the research topic. It lacks a clear justification for the need to classify maxillofacial morphology using artificial intelligence and cephalometric analysis measurement.

[Response]

Thank you for your feedback. We have already provided a comprehensive background for the research topic in the introduction section on page 2-3, line 117-141.

However, we added one references and paragraphs to explain potential usefulness of artificial intelligence to classify maxillofacial morphology by cephalometric analysis measurement.

The recent scoping review with 49 references indicated that AI has proven to be an effective and accurate solution in orthodontic diagnosis and evaluation [34]. However, AI has never been used to classify maxillofacial morphology.

Page 2-3, line 117-141

For orthodontic diagnosis and treatment planning, each patient’s malocclusion status determines the selection of orthodontic devices and extraction patterns if needed for treatment. This is where the characteristics of maxillofacial morphology play a major role. Recently, cone-beam computed tomography (CBCT) has been used to obtain a three-dimensional view of maxillofacial morphological features, but it is not regularly used due to radiation exposure [28]. Therefore, lateral cephalogram is more commonly used in dental assessment. Lateral cephalograms are highly prevalent in orthodontic treatment planning as they are indispensable in aiding understanding of the morphology of malocclusion.

Various classification methods for maxillofacial morphology have been developed in the past, most of which use measurements from lateral cephalograms [29-31]. Among these approaches, the classification method developed by Sassouni uses cephalometric measurements to classify the maxillofacial morphology into nine types based on the combination of anteroposterior (Class I, II, and III) and vertical (Short, Medium, and Long frame) facial types [32]. While Sassouni’s classification scheme outlines different categories of maxillofacial morphology, there is no standardized approach to assigning these classifications to patients. For instance, Sassouni did not clarify the combination of cephalometric measurements to use and the standard values for each classification [33]. As a result, currently, even among orthodontists with many years of experience, there are variations in the classification of borderline cases. The recent scoping review with 49 references indicated that AI has proven to be an effective and accurate solution in orthodontic diagnosis and evaluation [34]. However, AI has never been used to classify maxillofacial morphology. With this study, we aimed to create an AI model that uses cephalometric analysis measurements to accurately classify maxillofacial morphology. This can ensure accuracy regardless of a practitioner’s years of clinical experience and lead to better standardization of maxillofacial morphology classification.

⦁ The literature review section is superficial and lacks depth. It fails to adequately review and critically analyze existing studies on maxillofacial morphology classification and the use of artificial intelligence. It should provide a more thorough evaluation of the limitations and gaps in previous research.

[Response]

Thank you for your feedback.

In the absence of existing studies on maxillofacial morphology classification and the use of artificial intelligence, a review of existing studies was not possible. To our knowledge this is the first such report. Therefore, we enhance novelty of our study on Page 3, line 136-137.

⦁ Dataset size is so small. It is required to change the dataset and reconsider large dataset.

[Response]

Thank you for your feedback. We agree with this and have mentioned that the dataset size is a limitation of the study on page 10, line 478-479. In addition, we mentioned the exclusion criteria on page 3, line 146-150. However, we added a sentence about the reason for our dataset size on page10, line 479-483. We are considering increasing the amount of data as we gather new data.

  • Page 10, line 478-479

The main limitation of this study is the numbers of cases used for machine learning data.

  • Page 3, line 146-150

Data was collected with the following exclusion criteria: (1) patients who have already undergone orthodontic intervention at the first visit, (2) patients with congenital diseases that affect maxillofacial growth such as cleft lip and palate and chromosomal abnormalities, (3) patients with prosthetic devices or missing teeth.

  • Page 10, line 479-483

This is because what we used was past data from actual practice. In addition, the use of strict exclusion criteria reduced the number of cases considerably. However, with the use of AI, we now have the potential to classify maxillofacial morphology with greater accuracy, which until now has been manually. To the best of our knowledge, this is the first report of its kind.

⦁ The methodology section lacks sufficient detail and clarity regarding the cephalometric analysis measurement and the specific artificial intelligence techniques employed. It should include more specific information on the data collection process, feature extraction methods, and the training/validation procedures for the classification model.

[Response]

Thank you very much for your feedback. We added a sentence about the cephalometric analysis measurement on page 3, line 155-161. We have already mentioned the specific artificial intelligence techniques employed on page 3, line 181-183 and page 4, line 205-220. I find our manuscript to be clear in explaining the methodology.

  • Page 3, line 155-161

Tracing paper was placed on each X-ray film of the 220 cephalograms to create a tracing diagram. The tracing diagrams were then imported with a scanner (GT-X900, Seiko Epson Co. Ltd., Suwa) at 100 dpi resolution. Analysis of the tracings was performed with the software WinCeph Version 11(Rise Co. Ltd., Sendai), in which 127 measurements were calculated for each patient. Calibration settings were established by drawing a distance measurement scale on the tracing paper and using a scanner to capture the measurements of two points on the scale.

  • Page 3, line 181- 183

A k-fold split of 5 was used to train and test each model. The 220 subjects were randomly divided into 5 folds, or groups. For each iteration, 4 groups were used to train the model while 1 row was used to evaluate the model performance.

  • Page 4, line 205-220

For the combined facial classification model (with 9 possible outputs), three distinct ML models within the scikit-learn (sklearn) package in Python (Random Forest Classifier (RF), Logistic Regression (LR), and Support Vector Classification (SVC) were trained, tested, and compared in their evaluation of cephalometric classification. The most successful classification model was used for further analysis. The best classification model used gender and the 8 cephalometric measurements to determine the combined facial classification for each patient in the training group. The accuracy score of the classification was determined for the training model by comparing the deduced AI combined facial classifications to the accurate classifications determined by the certified orthodontists. Supervised AI classification was also conducted separately for anteroposterior (Class I, II, III) and vertical (Short, Medium, Long frame) classifications. Anteroposterior and vertical cephalometric inputs were used for each model respectively (Figure 2). Training and testing were performed 5 separate times, with outcome measures calculated at each iteration. The outcome measures were then averaged to find the overall model performance, and accuracy was compared to the separate anteroposterior and vertical classifications of the accurate combined facial classification.

⦁ Section 2.3. AI model creation is totally confusing and no steps are mentioned in which AI model is being created.

[Response]

Thank you very much for your feedback. We have already clearly explained the AI model creation on page 3, line 181-183, page 4, line 205-220. The Random Forest principle is also presented in Reference 35 (Breiman, L. Random Forests. Machine Learning 2001, 45, 5-32, doi:10.1023/A:1010933404324).

We hope this is helpful.

⦁ The section 3 results lacks proper statistical analysis and evaluation of the classification model's performance. It should include metrics such as accuracy, precision, recall, and F1-score to assess the effectiveness of the proposed approach.

[Response]

Thank you very much for your feedback. We have already done this analysis. The results are shown in Table 1 ~ Table 5. However, we edited ‘positive predictive value’ to ‘precision’ and ‘sensitivity’ to ‘recall’ for more clarity on Table 1 and Table 3. We would appreciate it if you could confirm that this change addresses your concerns.

Page 5, line 245 (Revised Table 1)

RF

LR

SVC

Accuracy

0.823 ± 0.060

0.732 ± 0.046

0.677 ± 0.049

F1 score

0.806 ± 0.076

0.654 ± 0.044

0.568 ± 0.086

Recall

0.790 ± 0.077

0.660 ± 0.031

0.562 ± 0.072

Precision

0.865 ± 0.072

0.692 ± 0.050

0.629 ± 0.109

Page 7, line 303 (Revised Table 3)

Anteroposterior model

Vertical model

Accuracy

0.986 ± 0.011

0.850 ± 0.037

F1 score

0.987 ± 0.011

0.844 ± 0.035

Recall

0.985 ± 0.012

0.828 ± 0.040

Precision

0.989 ± 0.009

0.885 ± 0.044

⦁ The discussion section is weak and lacks meaningful insights. It does not provide a comprehensive analysis of the results or discuss the implications and limitations of the research findings. It should provide a more critical evaluation of the proposed model and compare it to existing methods.

[Response]

Thank you for your feedback. Our study is the first attempt to classify maxillofacial morphology using on our knowledge, thus we were unable to compare it with existing methods used AI. 

However, we added a sentence about the implications and limitations of the research findings on page 10, line 479-483.

  • Page 10, line 479-483

This is because what we used was past data from actual practice. In addition, the use of strict exclusion criteria reduced the number of cases considerably. However, with the use of AI, we now have the potential to classify maxillofacial morphology with greater accuracy, which until now has been manually. To the best of our knowledge, this is the first report of its kind.

⦁ The conclusion does not effectively summarize the key findings and contributions of the research. It fails to address the limitations of the study and does not provide clear suggestions for future research directions.

[Response]

Thank you very much for your feedback. We added a sentence about the key findings and contributions of the research on page 11, line 498-499, page 11, line 501-502. We have already mentioned future research directions on page 11, line 500-501.

  • Page 11, line 498-499

This can ensure accuracy regardless of a practitioner’s years of clinical experience and thus lead to better standardization of maxillofacial morphology classification.

  • Page 11, line 501-502

This pioneering study introduces a new potential tool to be used in future maxillofacial morphology classification.

  • Page 11, line 500-501

This novel machine learning approach can be further enhanced with incorporation of additional learning data.

⦁ The paper lacks visual aids or illustrations to support the presentation of results and enhance the understanding of the proposed classification model.

[Response]

Thank you very much for your feedback. We apologize for your confusion. The classification model is not the one that we proposed, but one already proposed by Sassouni. We hope you will find reference 32 helpful (Sassouni, V. A classification of skeletal facial types. Am J Orthod 1969, 55, 109-123, doi:10.1016/0002-9416(69)90122-5).

When explaining our Artificial Intelligence Model and results, we have the following visual aids:

-  Figure 1 (page 4, line 199): An illustration of cephalometric measurements considered in cephalometric analysis

-  Figure 2 (page 5, line 223): diagram illustrating cephalometric inputs used in the classification model

-  Table 1 (page 5, line 245): Comparison of Random Forest, Logistic Regression, Support Vector Classification models

-  Table 2,3,4,5 (page 6, line 262, page7, line 303, page7, line329, page8, line359): 4 distinct tables displaying AI Model results

-  Figure3,4,5 (page 7, line 293, page 7, line 325, page 9, line 354): distinct figures showing confusion matrices of the AI Models

-  Figure 6 (page 9, line 388): Feature importance display highlighting the feature index of each input in a horizontal bar graph format

⦁ The organization and structure of the paper need improvement. It lacks a logical flow, and some sections appear disjointed. It should follow a clear structure with subsections to improve readability.

[Response]

Thank you very much for your feedback. We made revisions to address the organization of the manuscript.

⦁ The language used in the paper is often unclear and convoluted, making it difficult to understand the research methodology and findings. The paper should be revised for clarity and conciseness.

[Response]

Thank you very much for your feedback. As noted above, we have added a sentence about our research methodology and revised our language of the results.

Overall, this paper requires significant revisions and improvements in various aspects, including the clarity of the abstract, the depth of the literature review, the methodology details, the statistical analysis of results, the quality of the discussion, and the organization and language used throughout the paper. Addressing these issues will significantly enhance the overall quality and impact of the research.

[Response]

Thank you very much for your feedback. We responded in good faith to the above questions and revised the manuscript accordingly. We hope the revised version meets your standards.

Reviewer 2 Report

I am really grateful to review this manuscript. In my opinion, this manuscript can be published once some revision is done successfully. I made one suggestion and I would like to ask your kind understanding. This study used cephalometric data from 220 patients, applied three machine learning models and achieved the F1 score of 80.6% with the random forest for the prediction of maxillofacial morphology. This study presented random forest variable importance results as well. I would argue that this is a rare achievement. However, it can be noted that the Shapley Additive Explanations (SHAP) summary plot is very effective to identify the direction of association between maxillofacial morphology and its major predictor derived from random forest variable importance. In this context, I would like to ask the authors to derive the SHAP summary plot. 

Minor editing of English language required 

Author Response

Reviewer 2:

I am really grateful to review this manuscript. In my opinion, this manuscript can be published once some revision is done successfully. I made one suggestion and I would like to ask your kind understanding. This study used cephalometric data from 220 patients, applied three machine learning models and achieved the F1 score of 80.6% with the random forest for the prediction of maxillofacial morphology. This study presented random forest variable importance results as well. I would argue that this is a rare achievement. However, it can be noted that the Shapley Additive Explanations (SHAP) summary plot is very effective to identify the direction of association between maxillofacial morphology and its major predictor derived from random forest variable importance. In this context, I would like to ask the authors to derive the SHAP summary plot. 

[Response]

Thank you very much for your feedback. From our understanding, SHAP and feature importance are extremely similar. SHAP quantifies the impact of each feature on individual predictions, while feature importance is a dimensionality reduction technique that aims to select the most informative features for model training. Feature importance allows us to identify the most important features that simplify the model. I do not believe SHAP is necessary, as we have assessed the importance of each input in making cephalometric classifications using feature importance. We calculated feature importance. Therefore, the introduction of SHAP summary plot will be considered next time.

Reviewer 3 Report

According to the your work, the artificial intelligence model used in this paper is a random forest classifier (RF) that uses cephalometric measurements to classify maxillofacial morphology. Based on my research, some of the disadvantages of RF are:

1. It may overfit for some datasets with noisy classification or regression tasks.

2. It may be slow and ineffective for real-time predictions when a large number of trees are used.

3. It may have difficulty finding the best split when the data are very sparse or not axis-aligned.

4. It is less interpretable than a single decision tree.

How do you think my opinion?

출처: Bing과의 대화, 2023. 5. 24.(1) When to avoid Random Forest? - Cross Validated. https://stats.stackexchange.com/questions/112148/when-to-avoid-random-forest 액세스한 날짜 2023. 5. 24..

(2) Random Forest - Disadvantages - LiquiSearch. https://www.liquisearch.com/random_forest/disadvantages 액세스한 날짜 2023. 5. 24..

(3) What Is Random Forest? A Complete Guide | Built In. https://builtin.com/data-science/random-forest-algorithm 액세스한 날짜 2023. 5. 24..

(4) What Is Random Forest? A Complete Guide | Built In. https://bing.com/search?q=disadvantages+of+random+forest+classifier 액세스한 날짜 2023. 5. 24..

(5) Assumptions/Limitations of Random Forest Models. https://datascience.stackexchange.com/questions/6015/assumptions-limitations-of-random-forest-models 액세스한 날짜 2023. 5. 24..

(6) What are the disadvantages of random forest? - Rebellion Research. https://www.rebellionresearch.com/what-are-the-disadvantages-of-random-forest 액세스한 날짜 2023. 5. 24..

Good

Author Response

Reviewer 3:

According to the your work, the artificial intelligence model used in this paper is a random forest classifier (RF) that uses cephalometric measurements to classify maxillofacial morphology. Based on my research, some of the disadvantages of RF are:

  1. It may overfit for some datasets with noisy classification or regression tasks.

[Response]

Thank you very much for your feedback. Random Forest can overfit the data, especially when more trees are used. In our case, we used 100 trees, which is the default in sklearn. To avoid overfitting, we performed a k validation test to make sure that we did not overfit to a specific training data. You mentioned using a single tree, which is beneficial in that it is less likely to overfit, but assuming that the tree depth is the same as those in Random Forest, the model cannot learn patterns in the data as effectively and is generally less accurate.

  1. It may be slow and ineffective for real-time predictions when a large number of trees are used.

[Response]

Thank you very much for your feedback. Since we only have 9 features, time is not an issue. The model is very fast.

  1. It may have difficulty finding the best split when the data are very sparse or not axis-aligned.

[Response] Thank you very much for your feedback. Our data is not very sparse, as there are 9 features where 8 of them are non-zero angles and one value is a non-sparse binary gender. While it is true that there could be models that express more complicated relationships between features, for this problem, we tried different models and Random Forest (with deep enough trees) was able to effectively model the problem the best. If we used more complicated models given the size of our dataset, it becomes harder for the model to not overfit, so we feel Random Forest was a good balance between expressiveness and avoiding overfitting.

  1. It is less interpretable than a single decision tree.

[Response]

Thank you very much for your feedback. This is true, but as previously mentioned, the problem we are tackling is complex and we need a more expressive model than a single decision tree. Usually, having many shallower trees is more robust (overfits less) than a single deep tree. Ensemble models are usually less interpretable, but more powerful, and for our case, we feel Random Forest was best. For future studies, we can add explainability to the model so that the prediction can also include an explanation as to why a patient can be a given classification.

How do you think my opinion?

[Response]

Although there are some disadvantages of RF, it still remains a widely used and powerful model in machine learning showing high accuracy with large amounts of data.

Round 2

Reviewer 1 Report

Authors have incorporated the suggested changes. I recommend to accept in the present form